# Mechanical analysis of eight-legged temporary support structure under wave forces

**Weikuan Wang**[1,2], **Yan-zhao Yuan**[1] *

1 MSc Civil Engineering, University of Exeter, Exeter, England, 2 Lecturer College of Civil and Traffic Engineering, Henan University of Urban Construction, Pingdingshan, 467036, China

* yyz@hncj.edu.cn

**Data Availability Statement:** All relevant data are within the paper and its Supporting Information files.

**Funding:** The National Natural Science Foundation of China with grant number 52004081. Public

## Abstract

This paper presents a mechanical analysis of the foundation of a temporary offshore platform using a temporary embedded eight-leg support structure. The model is simulated using the finite element simulation software MIDAS-3D, with the modified RANS equation and Forchheimer saturated resistance model used to control the fluid. The stress analysis principle of the structure is simplified by the pile group theory. The stability of the eight-legged supporting structure is investigated under different embedding depths, pile diameters, wave periods, and amplitudes of the main piles. The results show that the eight-legged supporting structure can intercept and divert water flow, eliminating the impact of the water flow on the main piles during large waves. Additionally, as the diameter of the structure increases under the same wave conditions, the influence of the base volume and surface curvature gradually increases, deteriorating the stress environment of the main pile and decreasing the supporting effect of the eight-legged support structure. Numerical calculations of the seabed rock foundation of the eight-leg supporting structure show that the shallow pile foundation undergoes significant deformation, while the eight-leg supporting structure is still supported by the dead weight of the main pile.

## 1. Introduction

With the construction of offshore wind farms and drilling platforms, there is a growing trend towards using single pile foundations, particularly for wind turbines. However, during the construction process of single pile foundations, temporary stable structures are necessary. These structures provide stability and support during the construction phase until the permanent foundation is completed. (as shown in Fig 1). Compared to long-term buried offshore structures, shallow buried structures exhibit lower stability and safety due to their specific engineering characteristics. In particular, the effects of wind and waves have a significant impact on the stability of shallow buried offshore equipment. It is crucial to thoroughly consider and assess these factors to ensure the safe and reliable operation of such structures [1–6]. The vulnerability of single-pile offshore wind support structures under the combined action of wind, waves, and earthquakes was examined and discussed. The study revealed that the

Relations Program of Science and Technology of Henan Province (Industrial Field) Project with grant number 182102210221.The funders had no role in study design, data collection and analysis, decision to publish, or preparation of the manuscript.

**Competing interests:** The authors have declared that no competing interests exist.

simultaneous influence of wind and waves could lead to damage in the pile foundation structure. This highlights the importance of considering the combined effects of these environmental factors in the design and construction of offshore wind support structures to ensure their integrity and resilience [7]. Mo [8] analyzed the seismic response relationship of structures such as tripod and guide frame and found that wind and waves can cause high displacement differences at the bottom of the tower. Matlock [9] conducted a series of tests on steel pipe piles embedded in clay, including horizontal static load and cyclic load tests. The objective was to obtain the P-S curve for static loading and investigate the changes in the soil's P-S curve after cyclic loading. The objective of the study was to evaluate the behavior of steel pipe piles in clay under various loading conditions. Specifically, the focus was on analyzing the load-settlement characteristics of the piles and investigating the impact of cyclic loading on the soil's response. Through conducting these tests, valuable insights were gained regarding the performance of steel pipe piles and their interaction with the surrounding clay soil. The obtained results contribute to a better understanding of the behavior and design considerations of steel pipe piles in similar geological conditions, aiding in the development of more effective and reliable foundation solutions.Due to the complexity of the factors to be considered in theoretical calculations, some scholars have introduced numerical simulation to solve such problems. For example,Jeng [10] applied the Navier-Stokes equation to determine the boundary conditions of the soil under wave action, and a boundary layer approximation was used to solve the velocity and pressure fields. Meanwhile, He [11] focused on numerical modeling of wave interactions with submerged porous structures, including reef breakwaters and low-crested structures. The study aimed to examine the influence of different dimensionless parameters on wave transmission. To achieve this, numerical analysis was conducted, considering variations in the crest width of the structure as well as different wave parameters [12–20]. By extending the analysis to these different scenarios, the study sought to understand how these factors impact wave transmission. The results obtained from this investigation contribute to a deeper understanding of the relationship between dimensionless parameters and wave transmission, providing valuable insights for the design and optimization of structures subjected to wave forces.

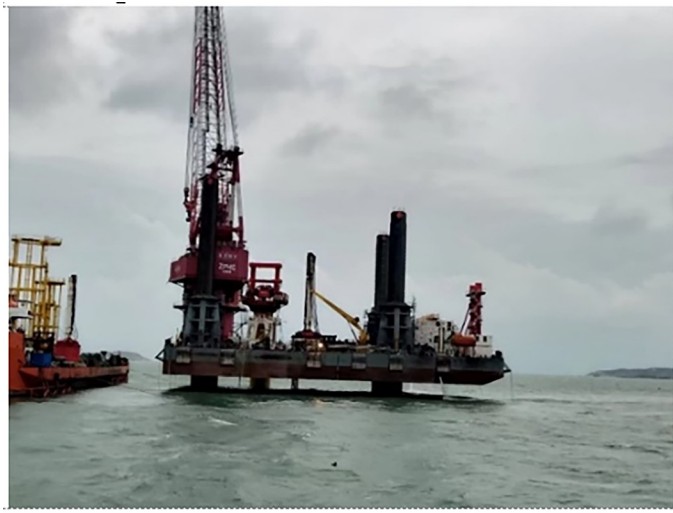

a) Four-leg cargo platform

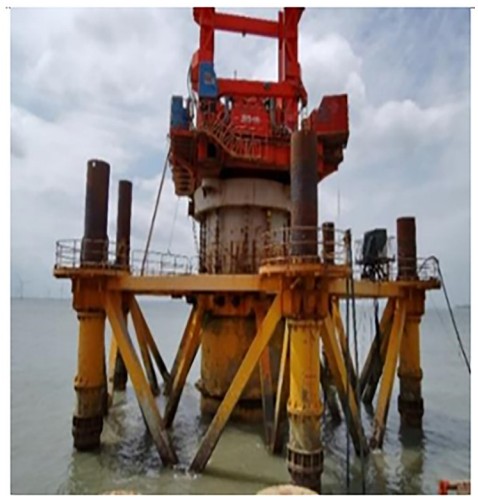

b) Eight-legged drilling platform

**Fig 1. The large diameter offshore drilling platform.**

This paper focuses on the stability analysis of a shallow buried eight-legged structure under wave action. The study utilizes the finite element simulation software MIDAS-3D to investigate the influence of waves on the structure. By employing pile group theory and the modified RANS equation, the stress distribution of the structure and underlying soil is analyzed and verified using relevant literature data. Additionally, relevant models are employed to examine the deformation characteristics of the structure under different wave conditions, pipe diameters, embedding depths, and soil structure displacements. Based on these deformation characteristics, the stability of the shallow buried eight-legged offshore structure is thoroughly discussed and analyzed. The findings of this study provide a valuable theoretical basis for further research on the stability of offshore temporary work platforms.

## 2 The numerical model

In this study, MIDAS-3D numerical simulation software is used to perform numerical modeling and analysis on the shallow and simple eight-legged stability structure of this project (as shown in Fig 2) [21–28]. According to the wave conditions in the South China Sea, the depth of the seabed is set to 10 meters and the height of the upper water layer is set to 20 meters. Therefore, based on the model size, the grid size in this study is set to 0.2m × 0.2m. The dimensions of the eight supports are cylindrical steel with a height of 22 meters and a diameter of 0.5 meters. The central part is the main pile foundation structure, which has a diameter of 10 meters and a height of 25 meters. The structure extends into the rocky seabed at a depth of dn, and dw represents water depth; ds represents the thickness of the seabed; $H_0$ represents the initial wave height; L represents the wave length; D represents the diameter of a single drilled casing; e represents the relative depth, which is the ratio of the depth of the structure extending into the rocky seabed dm to the thickness of the seabed ds. Since the ratio of the width of the structure to the numerical simulation model is less than 0.2, the size effect of the numerical model can be ignored, and accurate calculations can be performed on the relevant model.

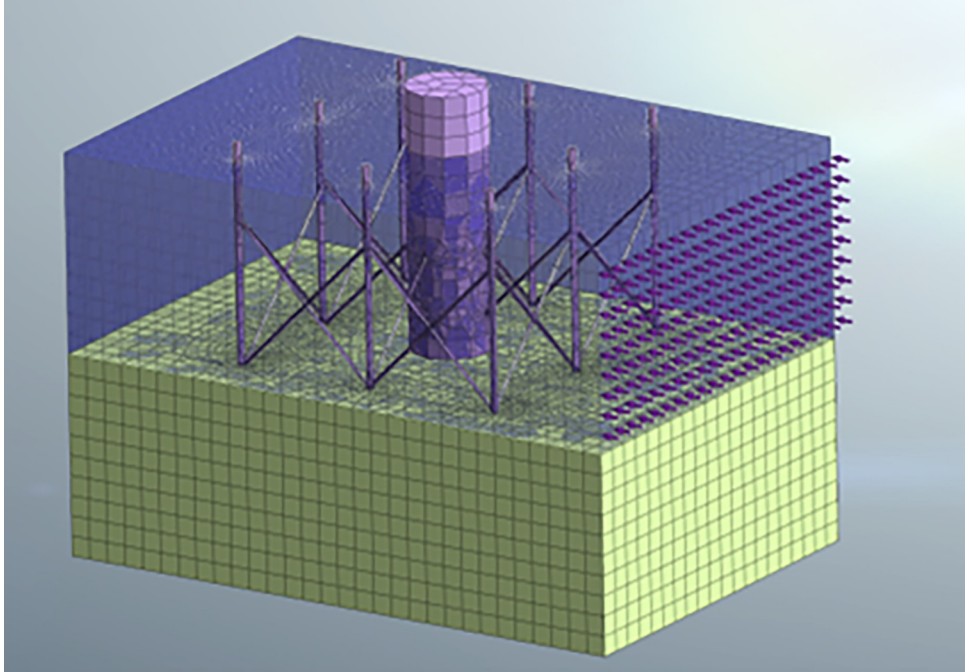

**Fig 2. Schematic diagram of numerical calculation of long-span particles on the rocky seabed.**

## 2.1 The governing equations

**2.1.1 The fluid motion control.** In this paper, the interaction forces between fluid and pile under wave action are analyzed based on the RANS equation. With the support of the k-epsilon equation [12], the following formula analysis is carried out for the wave action forces:

$$\frac{\partial \langle u_i \rangle}{\partial x_i} = 0 \tag{1}$$

$$\frac{\partial \rho \langle u_i \rangle}{\partial t} + \frac{\partial \rho \langle u_i \rangle \langle u_j \rangle}{\partial x_j} = -\frac{\partial \langle p \rangle}{\partial x_i} + \frac{\partial}{\partial x_j}\left[\mu\left(\frac{\partial \langle u_i \rangle}{\partial x_j} + \frac{\partial \langle u_j \rangle}{\partial x_i}\right)\right] + \frac{\partial}{\partial x_j}\left(-\rho \langle u_i' u_j' \rangle\right) + \rho g_i - F_d u_i \tag{2}$$

Where $x_i$ is cartea sian coordinate; $\langle u_i \rangle$ is the overall average velocity component; this time $\rho$ is fluid density; $\langle p \rangle$ is fluid pressure; $\mu$ is dynamic viscosity; $g_i$ is the acceleration of gravity; $-F_d u_i$ is the resistance action term of porous media to water flow, that is, drag force. $-\rho \langle u_i' u_j' \rangle$ is Reynolds stress term, and k-ε turbulence equation can be used to simulate turbulent flow at high Reynolds number.

Considering the influence of fluid viscosity on the pressure distribution around the pile, the relationship between the pressure and the velocity gradient can be described by the following equation:

$$-\rho \langle u_i' u_j' \rangle = \mu_t\left(\frac{\partial \langle u_i \rangle}{\partial x_j} + \frac{\partial \langle u_j \rangle}{\partial x_i}\right) - \frac{2}{3}\rho \delta_{ij}\kappa \tag{3}$$

Where $\mu_t$ is turbulent viscosity; $\kappa$ is turbulence kinetic energy; $\delta_{ij}$ is the Cronech symbol.

Substituting Formula (3) into Formula (2), we can get

$$\frac{\partial \rho \langle u_i \rangle}{\partial t} + \frac{\partial \rho \langle u_i u_j \rangle}{\partial x_j} = -\frac{\partial}{\partial x_i}\left[\langle p \rangle + \frac{2}{3}\rho\kappa\right] + \frac{\partial}{\partial x_j}\left[\mu_{\text{eff}}\left(\frac{\partial \langle u_i \rangle}{\partial x_j} + \frac{\partial \langle u_j \rangle}{\partial x_i}\right)\right] \tag{4}$$

Where $\mu_{\text{eff}} = \mu + \mu_t$ is the total effective viscosity

In numerical analysis, the standard equation of $\kappa$-ε turbulence is:

$$\frac{\partial \rho\kappa}{\partial t} + \frac{\partial \rho \langle u_j \rangle \kappa}{\partial x_j} = \frac{\partial}{\partial x_j}\left[\left(\mu + \frac{\mu_t}{\sigma_K}\right)\frac{\partial \kappa}{\partial x_j}\right] + \rho P_K - \rho\varepsilon \tag{5}$$

$$\frac{\partial \rho\varepsilon}{\partial t} + \frac{\partial \rho \langle u_j \rangle \varepsilon}{\partial x_j} = \frac{\partial}{\partial x_j}\left[\left(\mu + \frac{\mu_t}{\sigma_\varepsilon}\right)\frac{\partial \varepsilon}{\partial x_j}\right] + \frac{\varepsilon}{\kappa}(C_{\varepsilon1}\rho P_K - C_{\varepsilon2}\rho\varepsilon) \tag{6}$$

$$\mu_t = \rho C_\mu \frac{\kappa^2}{\varepsilon} \tag{7}$$

$$P_K = \frac{\mu_t}{\rho}\left(\frac{\partial \langle u_i \rangle}{\partial x_j} + \frac{\partial \langle u_j \rangle}{\partial x_i}\right)\frac{\partial \langle u_i \rangle}{\partial x_j} \tag{8}$$

where $\kappa$ is turbulence kinetic energy; $\varepsilon$ is the dissipation rate of turbulence kinetic energy; Parameters $C_\mu$, $\delta_k$, $\delta_\varepsilon$, $C_{\varepsilon1}$ and $C_{\varepsilon2}$ [12] satisfy: $C_\mu = 0.09$, $\delta_k = 1.00$, $\delta_\varepsilon = 1.30$, $C_{\varepsilon1} = 1.44$ and $C_{\varepsilon2} = 1.92$.

**2.1.2 Effect of pile group under horizontal load.** According to the calculation principles in Midas-3D software, the nonlinear p-y curves are mostly obtained through indoor tests on single piles. However, in actual designs, there are often pile groups, and when the distance between piles exceeds a certain value, the horizontal resistance of single piles tends to decrease. To account for this effect, Kantardgi [5] introduced a reduction coefficient as shown in Fig 3.

In addition, when the distance between piles is small, elastic interaction between piles occurs, and the reduction coefficient is defined as shown in Fig 4. In this case, the ultimate horizontal bearing capacity of the pile group is equal to the sum of the ultimate horizontal bearing capacity of each single pile. However, due to the influence of the pile group effect, the bearing capacity of the pile group will be smaller than the sum of the bearing capacity of each single pile, as shown in Fig 5. Therefore, the method proposed by Brown [26] to reduce the bearing capacity of a single pile should be applied.

There are various factors that can affect the pile group effect in geotechnical engineering. However, in this study, we only consider the effect of pile spacing. The impact of horizontal pile spacing on the reduction coefficient will be discussed in sections 2.1.3 and 2.1.4, while the influence of oblique spacing on the reduction coefficient will be explained in section 2.1.5.

**2.1.3 Influence of transverse pile spacing on reduction coefficient.** Several studies, such as Cox [28], Wang [29], and Lieng [30], have investigated the influence of lateral pile spacing on the pile group under horizontal load. As shown in Fig 6, the reduction factor can be obtained by the relationship with the b/s curve, where s represents the vertical center distance between piles and b represents the pile diameter.

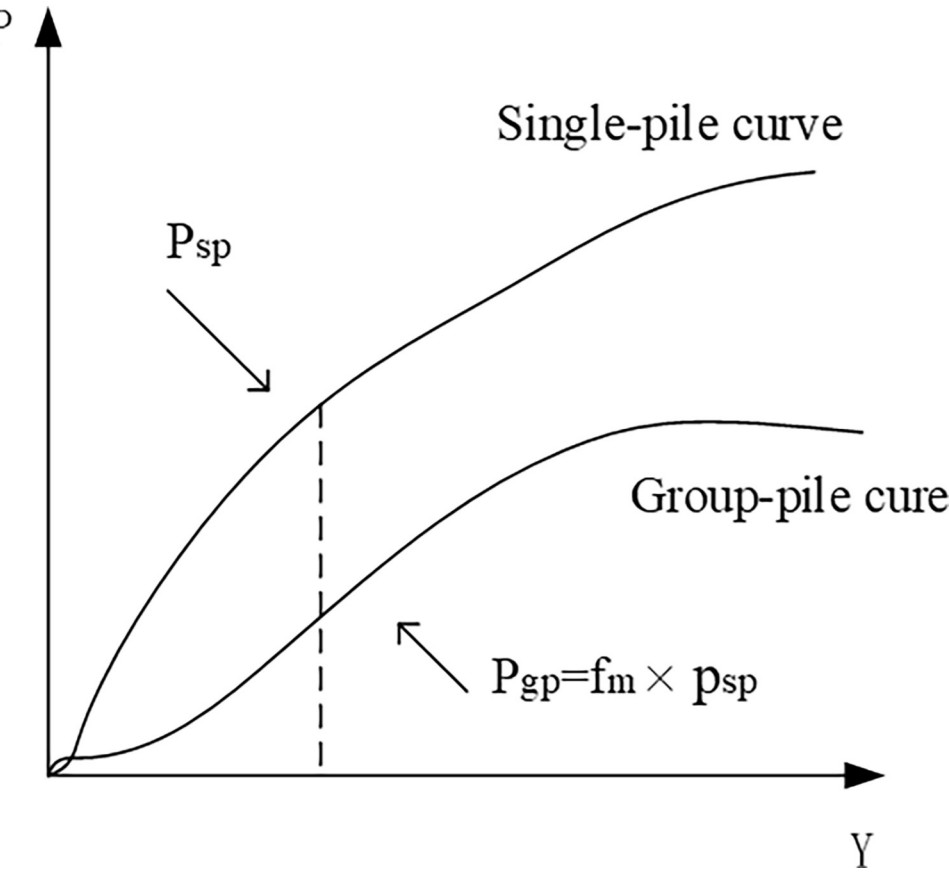

**Fig 3. Definition of FM.**

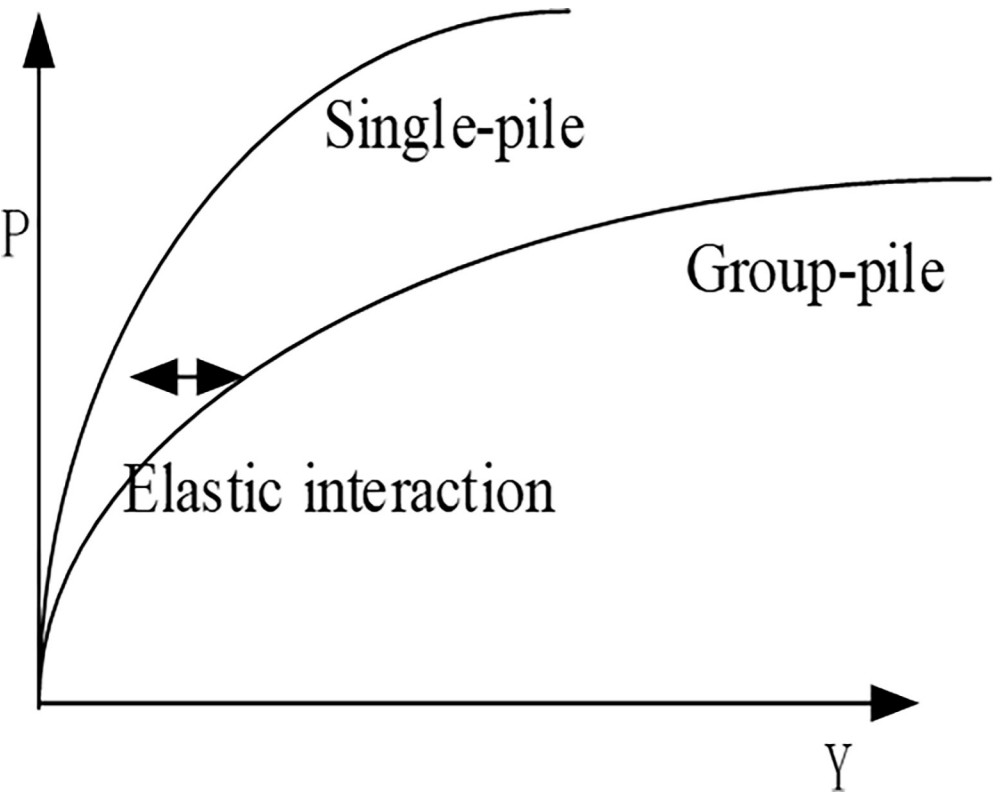

**Fig 4. Effect of pile group.**

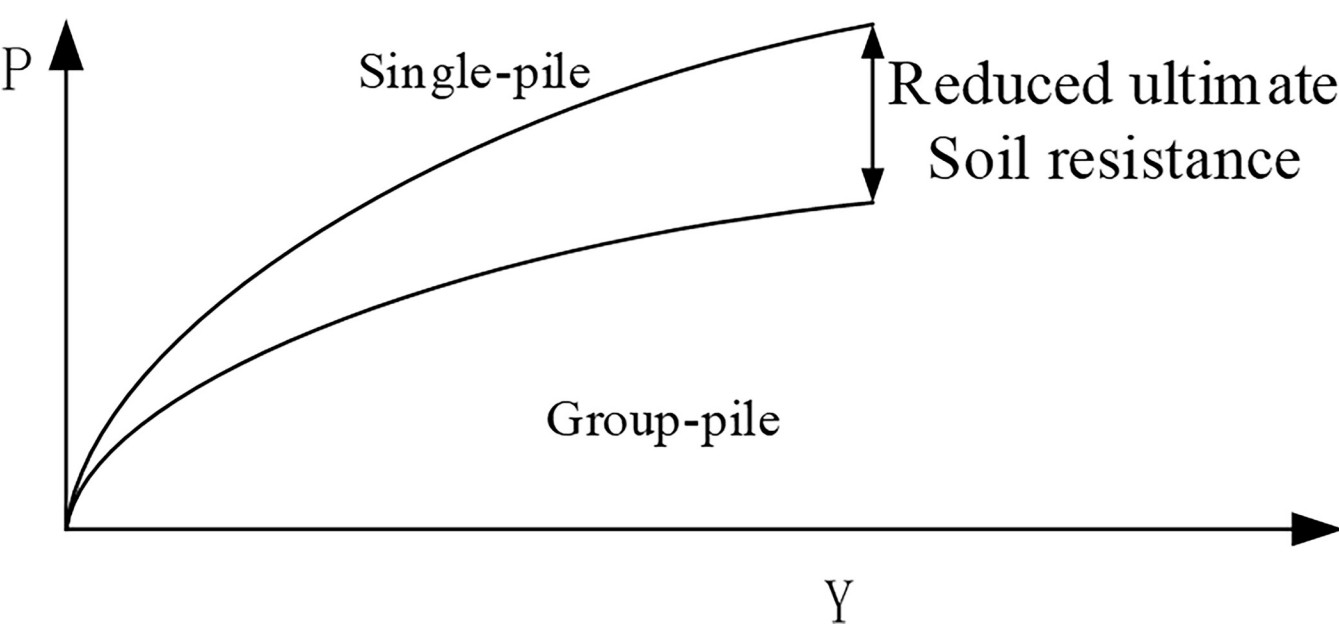

**Fig 5. The pile group effect leads to the reduction of the bearing capacity of single pile [27].**

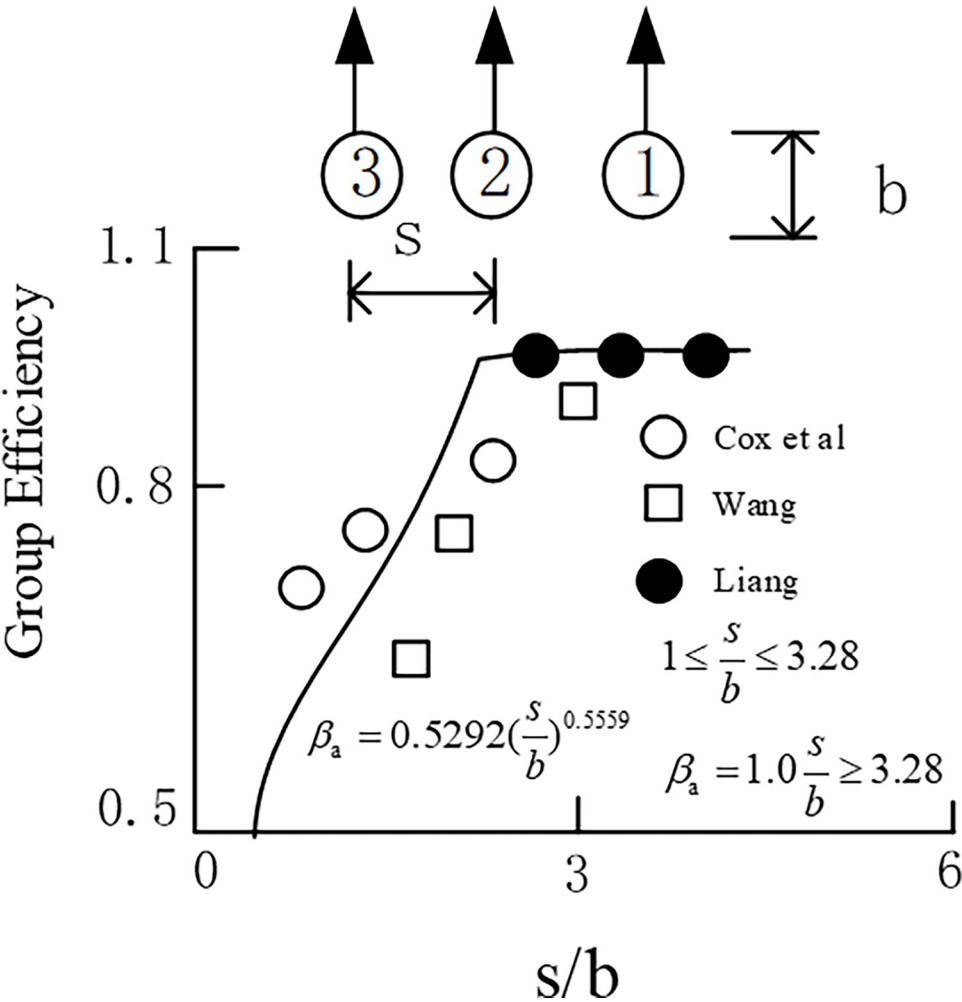

**Fig 6. Influence of reduction factor of lateral spacing of piles.**

From the figure, it can be observed that when the pile diameter is equal to the pile spacing, the ultimate bearing capacity decreases by 50%. However, when the distance between piles is greater than three times the diameter, the ultimate bearing capacity will not decrease. The reduction coefficient function is independent of the soil type.

**2.1.4 Influence of longitudinal pile spacing on reduction factor.** The longitudinal spacing of piles in the direction of load has a relatively complex influence on the reduction coefficient. Dunnavant [31] divided piles into front and rear piles, as shown in Fig 6. Pile A and No. 1 are the front piles of No. 2 and No. 3, and No. 2 is the front pile of No. 3. The reduction factor does not consider the characteristics of the soil. The influence of the longitudinal pile spacing of the first pile on the reduction coefficient is shown in Fig 7.

As shown in Fig 8, No.3 pile is the rear pile of No.1 pile and No.2 pile, and No.2 pile is the rear pile of No.1 pile. The influence of longitudinal pile spacing on the reduction coefficient of the rear pile is shown in Fig 8. It can be seen from the figure that when s/b is not less than 6, the reduction coefficient of the rear pile is almost unaffected.

**2.1.5 Influence of pile oblique spacing on reduction coefficient.** As shown in Fig 9 track, with the axis of the center and the load direction Angle, can use Fig 9 computation reduction factor a, use Figs 6 and 7 rainfall distribution on formula 10–12 computation

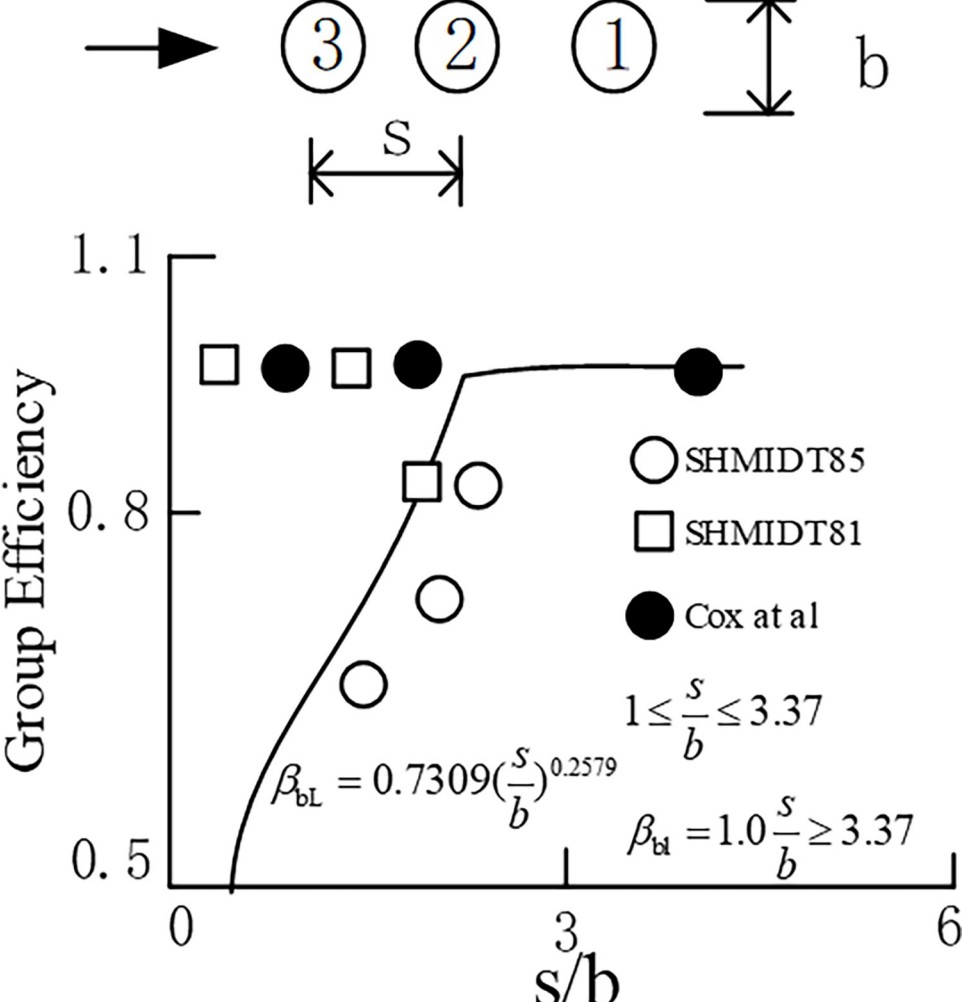

**Fig 7. Influence of longitudinal pile spacing of first pile on the reduction coefficient.**

reduction factor b, then use the following formula reduction factor.

$$\beta_s = (\beta_b^2 \cos^2\phi + \beta_a^2 \sin^2\phi)^{\frac{1}{2}} \qquad (9)$$

Where -The included angle between the load direction and the line connecting the pile center.

## 3 The model validation

To study the accuracy of the numerical simulation model, this paper quantitatively analyzes the results of the simulation based on previous experimental data of shallow-buried offshore platforms. A comparative analysis is carried out on the stress and deformation of pile foundation structures and the underlying soil layer under different wave forces.

### 3.1 The accuracy verification of wave impact on the eight-legged structure

In order to investigate the accuracy of the numerical simulation model, quantitative analysis was conducted in this study based on experimental data from previous shallow buried offshore

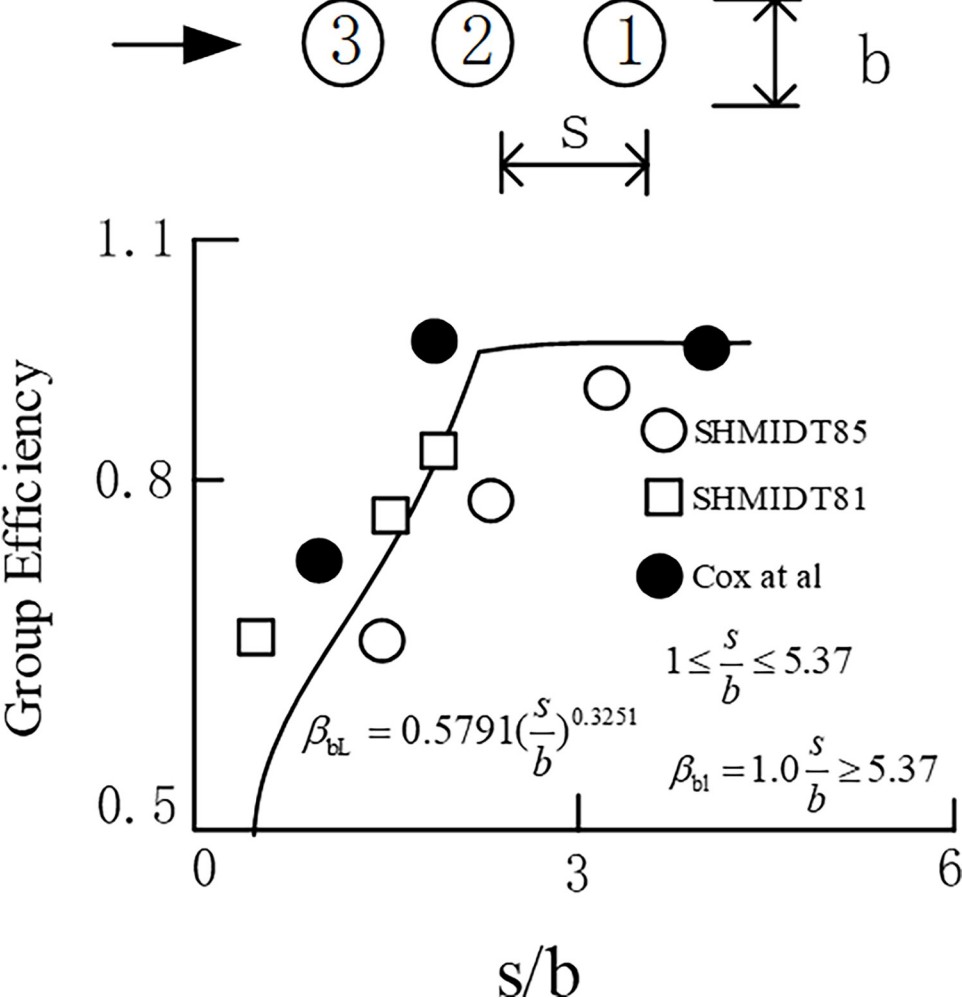

**Fig 8. Influence of longitudinal spacing of rear piles on discount coefficient.**

platform tests, comparing the force and deformation of the pile structures under different wave loads and soil conditions. Mo et al. [18] designed a series of flume tests to explore the interaction between waves and single piles. The tests were conducted in a 309 m long, 5 m wide and 7 m deep water tank, with a single pile fixed at the bottom of the horizontal flume 111 m away from the wave generator. Several wave gauges were arranged at different positions around the structure, and data from three typical locations (front, side, and back) were compared. The wave parameters were as follows: water depth dw = 4.76 m, wave period T = 4 s, wave height $H_0$ = 1.2 m, and single pile diameter D = 0.7 m.

Fig 10A) shows the stress distribution of the pile structure under the lowest wave amplitude, while Fig 10B) shows the stress distribution under the highest wave amplitude. Compared with the stress distribution under the lowest wave amplitude, the stress state of the octagonal column at the main periphery is higher at the front of the pile ($WG_1$), especially in the octagonal pile structure, where the vortex effect of the water flow leads to higher stress on the left front pole. The stress of waves is the highest in the trough. When the wave reaches the peak position, the force on the pile is the highest, and the $WG_1$ of the main stress cylinder is the largest, with a maximum value of 1.534 kN/m². The $WG_3$ of the main enclosure cylinder is lower, and the wave state changes greatly.

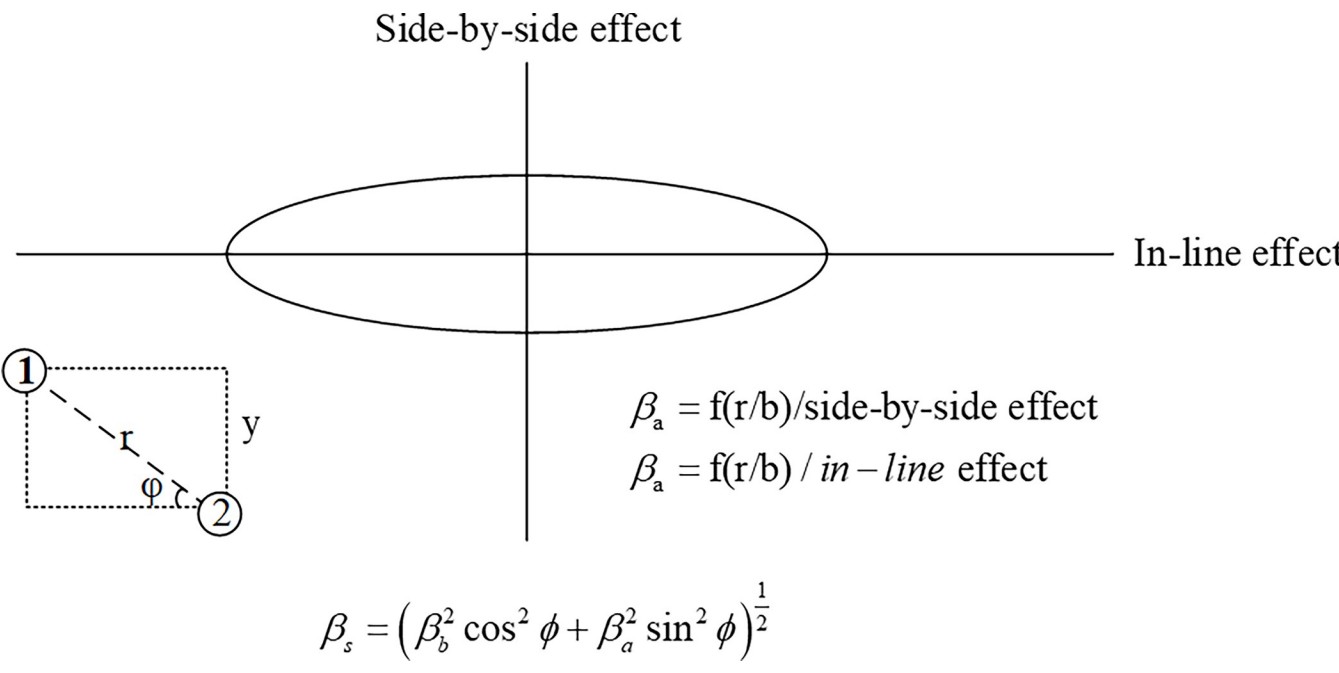

**Fig 9. Influence of the spacing of inclined piles on discount coefficient.**

Fig 10C) shows the time history curve of the relative wave height, where the dimensionless wave height η is calculated using η max and the experimental data of the reference wave gauge ($WG_{ref}$). The results show that the numerical model can reasonably simulate the wave flow around the lining and provide reliable analysis results for the follow-up study of wave forces on structures under wave action. The measured data are consistent with the numerical results, and some characteristic ripples are well simulated, such as noticeable ripples in the experimental and numerical results of the wave height at the front end ($WG_1$) when the wave trough passes through the drilling casing. The influence of the form on the wavefront can be ignored because the reference wave gauge ($WG_{ref}$) is far away from the structure.

### 3.2 The accuracy verification of the impact of waves on seabed bearing layer

In order to investigate the impact of rock seabed characteristics on wave forces on structures, a three-dimensional coupling model of wave-seabed-structure interaction was developed, taking into account both the rigidity (n = 0) and elasticity (n = 0.5) of the seabed. To verify the accuracy of the wave force calculation and the importance of considering seabed characteristics, numerical results of wave force FX along the wave propagation direction of liners under different seabed conditions were compared with MacCamy [32]'s theoretical results.

The numerical software analyzed the stress and displacement nephogram of the rock foundation, particularly the mechanics and displacement deformation laws of the shallow eight-legged pile foundation, as shown in Fig 11. The analysis along the wave direction revealed that there is almost no stress in the rock mass before the pile is buried in the hole, while the latter shows the diffusion of the step force, forming a bubble-like distributed stress distribution behind the main pile. Referring to Fig 4B, the pore of the rock foundation under the pile has caused serious deformation, and the maximum displacement of rock foundation at the front leg is 8.5.

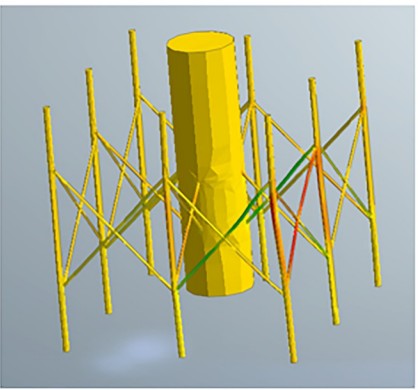

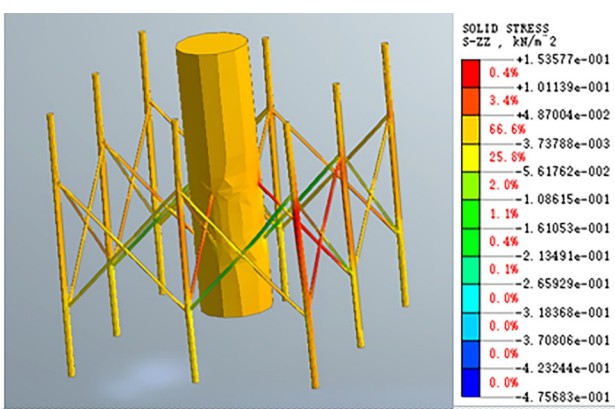

a）Force Distribution Diagram of Members during the Trough Period

b）Force Distribution Diagram of Members during the Crest Period

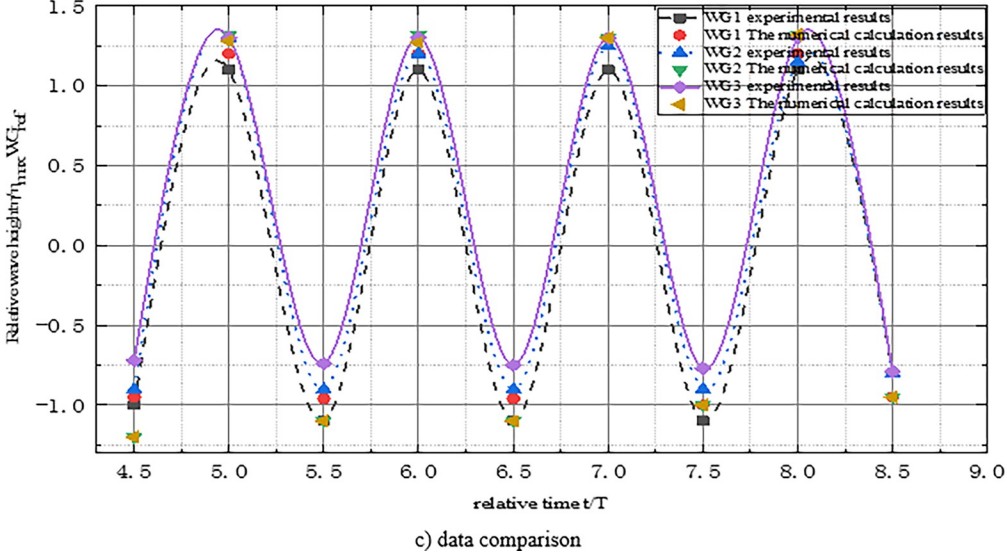

c) data comparison

**Fig 10. Comparison of relative wave elevation time-history curves.**

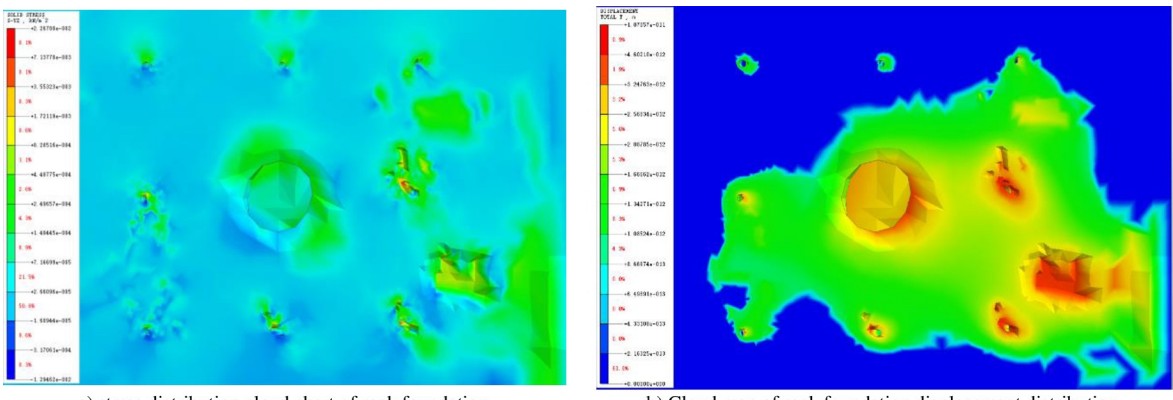

a) stress distribution cloud chart of rock foundation.

b) Cloud map of rock foundation displacement distribution.

**Fig 11. Cloud image of numerical simulation model.**

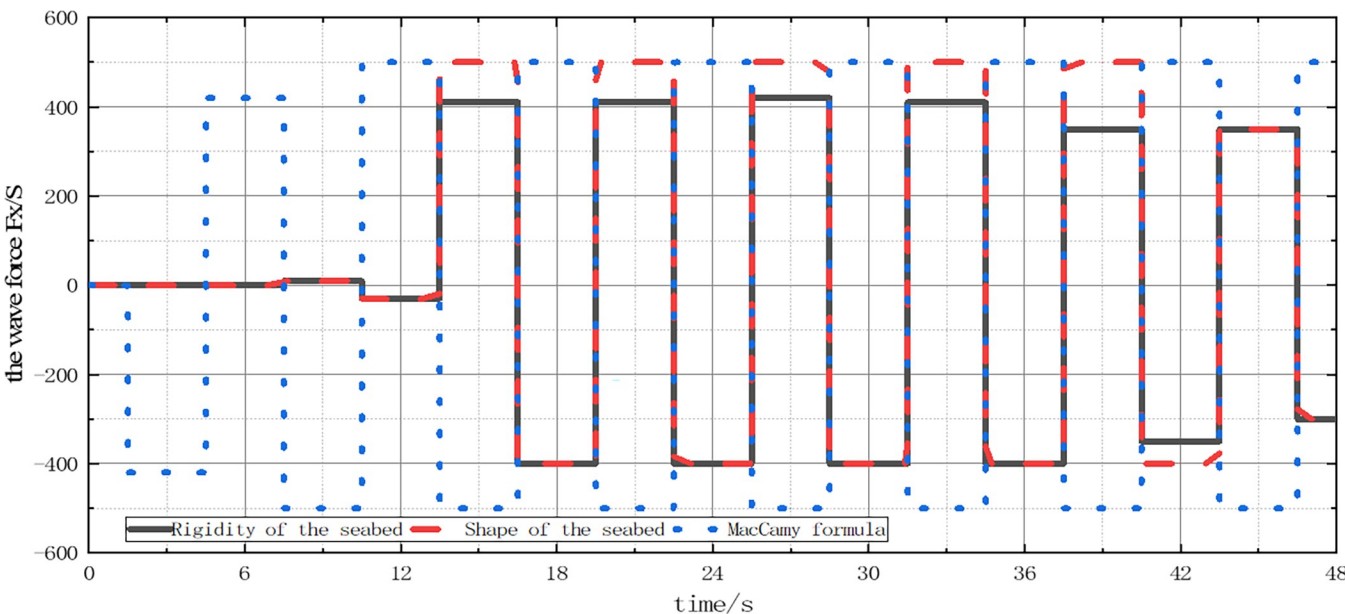

**Fig 12. Time history curve of wave force $F_X$ on a single pile.**

Fig 12 shows the FX time history curves under different seabed conditions. The comparison results indicate that the rigid seabed numerical analysis results agree with the MacCamy theoretical developments at the peak position, but there is a slight deviation at the valley position. This is mainly because the MacCamy theoretical formula does not consider the lateral force caused by vortex discharge behind the structure, which makes the amplitude of MacCamy's hypothetical results at the valley position larger than the numerical analysis results of the rigid seabed. Additionally, the maximum value of FX in the elastic seabed is about 35% higher than that in the rigid seabed. This highlights the accuracy of the three-dimensional coupling model in calculating the change of wave force on the structure under rigid seabed conditions, and the significance of seabed characteristics on the numerical value of wave force.

According to the comparison of numerical simulation analysis and actual data, it is known that the numerical analysis results can accurately reflect the actual stress situation of the soil. During the process of increasing wave forces, the deformation of the windward wave-facing side of the eight-legged structure reached 0.2 meters. As for the embedded soil at the bottom, the bottom embedding of the four structural bodies on the windward side experienced greater force, and the maximum deformation occurred in the middle embedding structure. The main pile structure in the middle plays a crucial role in the overall stability, especially when compared with the bottom deformation of the structures before and after the main pile. It is evident that the main pile bears most of the wave forces.

## 4 Discussion

### 4.1 The impact of wave intensity

This paper provides a detailed analysis of the structural forces of an eight-legged shallow-buried structure under different wave intensities, in accordance with the "Hydrographic Code for Ports and Waterways (JTS 145–2015) [33]" classification of wave forces. Three different wave levels with different wave heights, periods T, wave steepness ($H_0/L$), and relative water depth ($d_W/L$) were set up to study the effects of these parameters on wave forces on the cushion.

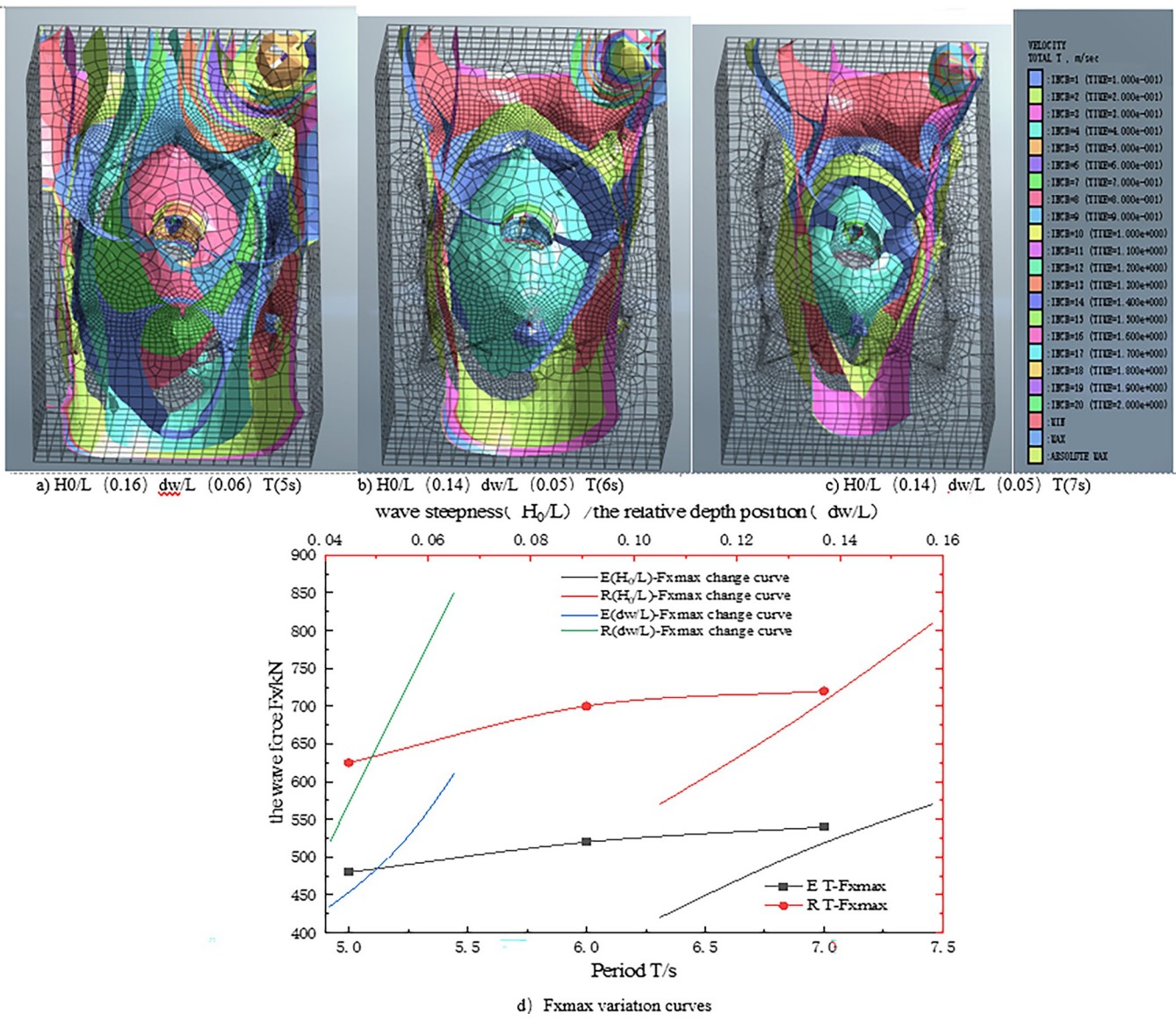

**Fig 13. Influence law of fluctuation parameters.**

Fig 13A–13C illustrate the three wave parameters ($H_0$/L (0.16), $d_w$/L (0.06) T (5s); $H_0$/L (0.14), $d_w$/L (0.05) T (6s); $H_0$/L (0.14).$d_W$/L (0.05) T (7s)), respectively. Fig 13A shows that when the wave force passes through the pile foundation of the eight-legged supporting structure, obvious tensile cracks appear on the back surface, while the wave force on the surface is too large due to the wave height, and then it is completely opened. Fig 13B shows that with the decrease of wave force, the wave is greatly restricted when passing through the eight-legged support structure, and the internal structure is rapidly compressed, resulting in severe turbulence. Fig 13C shows that when the wave force decreases to a certain extent, the wave force inside the eight-legged support structure increases sharply, and the water flow through the eight-legged structure is mostly affected by the eight-legged support pile, forming a relatively gentle turbulence structure.

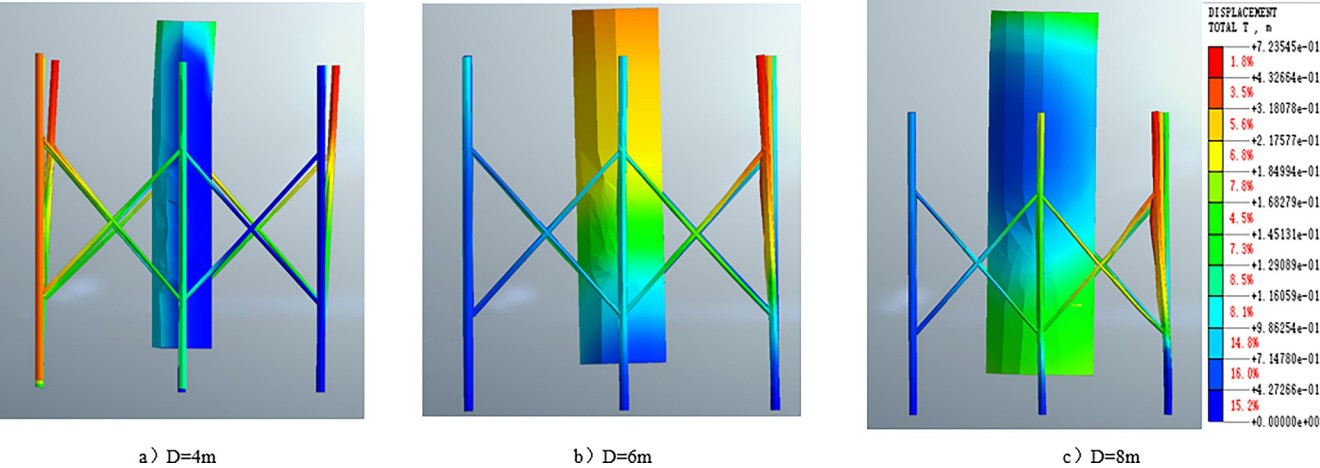

a）D=4m b）D=6m c）D=8m

**Fig 14. Nephogram of pile foundation displacement analysis of different diameters.**

Fig 13D shows the $FX_{max}$ curves under different wave periods T, wave steepness ($H_0$ /L), and relative water depth ($d_w$ /L). The results show that the $FX_{max}$ value increases linearly with the increase of wave parameters, but the increased range is different. The maximum value of wave force $FX_{max}$ is sensitive to wave height and water depth, indicating that the liners extending into the elastic seabed are subjected to greater wave forces under the action of big waves and long periods. This is because the wave energy is significant, and the projection area of the transverse structure to the wave propagation direction is substantial, resulting in a large reference volume, which makes the force borne by the cushion large. Additionally, the value of elastic seabed (E), $FX_{max}$ is greater than that of the rigid seabed (R). By comparing the analysis results, the importance of considering the seabed's strength in shallow-buried ocean engineering design is emphasized again.

## 4.2 The influence of main pile structural parameters

Since the stability of the eight-legged structure system is largely determined by the main pile structure, the diameter, minimum buried depth, and displacement of the foundation are particularly important for the design of the main pile. Therefore, this section uses numerical simulation to analyze the structure. It is beneficial to study the influence of cross-sectional diameter (d), shape, buried depth (e), and construction lowering speed (v) on the dynamic response of structures, which can help ensure structural safety.

**4.2.1 Stability of main pile structure with different sizes.** In order to investigate the influence of single pile diameter on wave force considering the porous characteristics of the seabed, the study increased the single pile diameter from 4m to 8m while keeping the same wave and seabed conditions. Fig 14A–14C show the contour plots of pile displacement with diameters of d = 4m, 6m, and 8m under wave action. It can be observed that as the pile diameter increases, the surrounding eight-legged pile structure gradually becomes stressed and tends to be stable. The maximum displacement occurs in the upper part of the first row of eight-legged piles along the wave direction when the maximum displacement at d = 4m decreases from 0.072m to d = 8m. When the main pile diameter is 4m, the main pile is deformed forward, but for pile diameters of 6m and 8m, their rigidity is relatively high, which helps maintain the stability of the octopus piles.

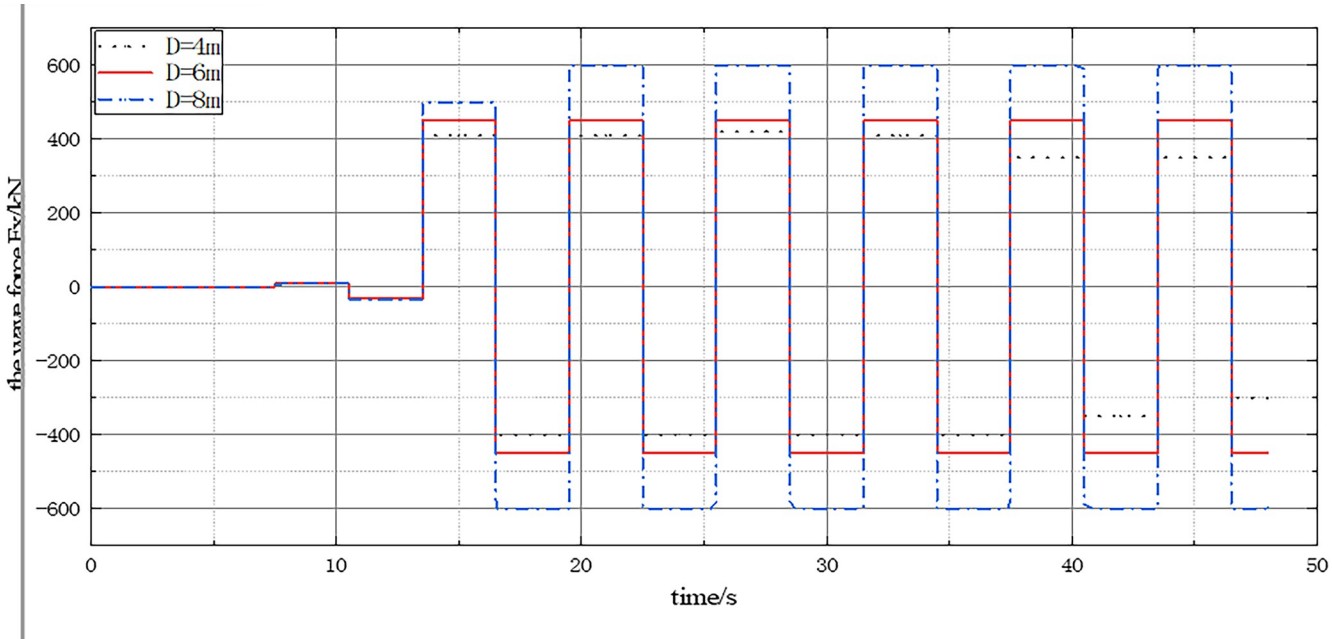

**Fig 15. The time history curve of FX under different pile foundation diameter D.**

Fig 15 presents the time history curve of wave force FX on a single pile. It shows that the wave force FX value oscillates with the wave period t and the amplitude remains stable when the wave is stable. Moreover, the amplitude of wave force FX on a single pile increases with the diameter of the pile. This is because the larger the diameter of a single pile, the larger the lateral stress area of the structure under the action of waves. In practical engineering, the diameter of systems can be determined using traditional calculation methods under the condition of a highly permeable seabed. Therefore, understanding the influence of single pile diameter on wave force is crucial for ensuring the structural safety of shallow-buried ocean engineering designs.

**4.2.2 The influence of different embedment depths on the main pile structure.** The simulation results presented in this paper indicate that the buried depth of the octopus pile

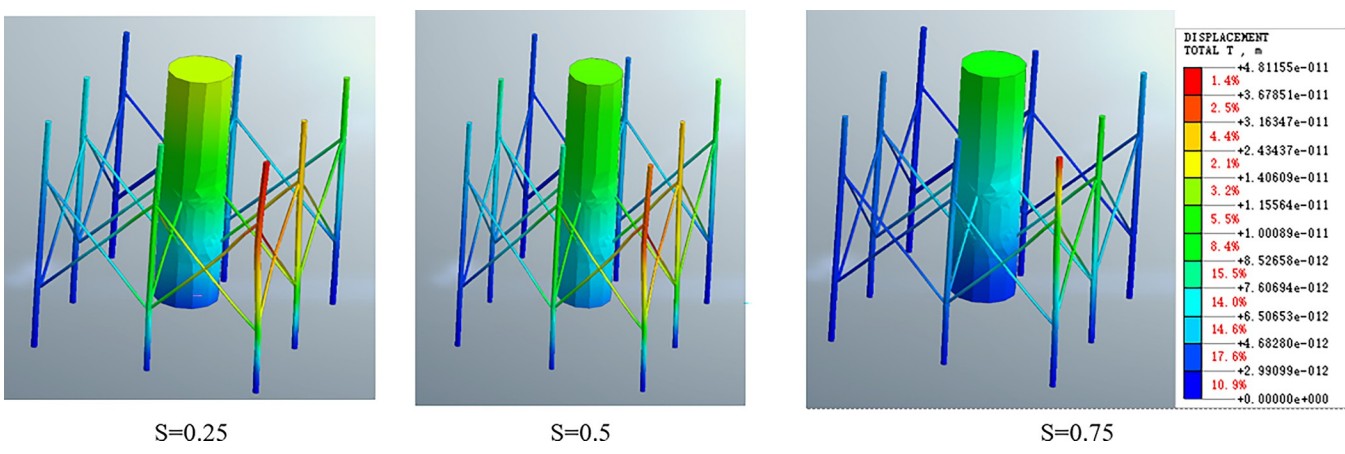

**Fig 16. Cloud image of deformation of eight-leg supporting structure under different buried depths.**

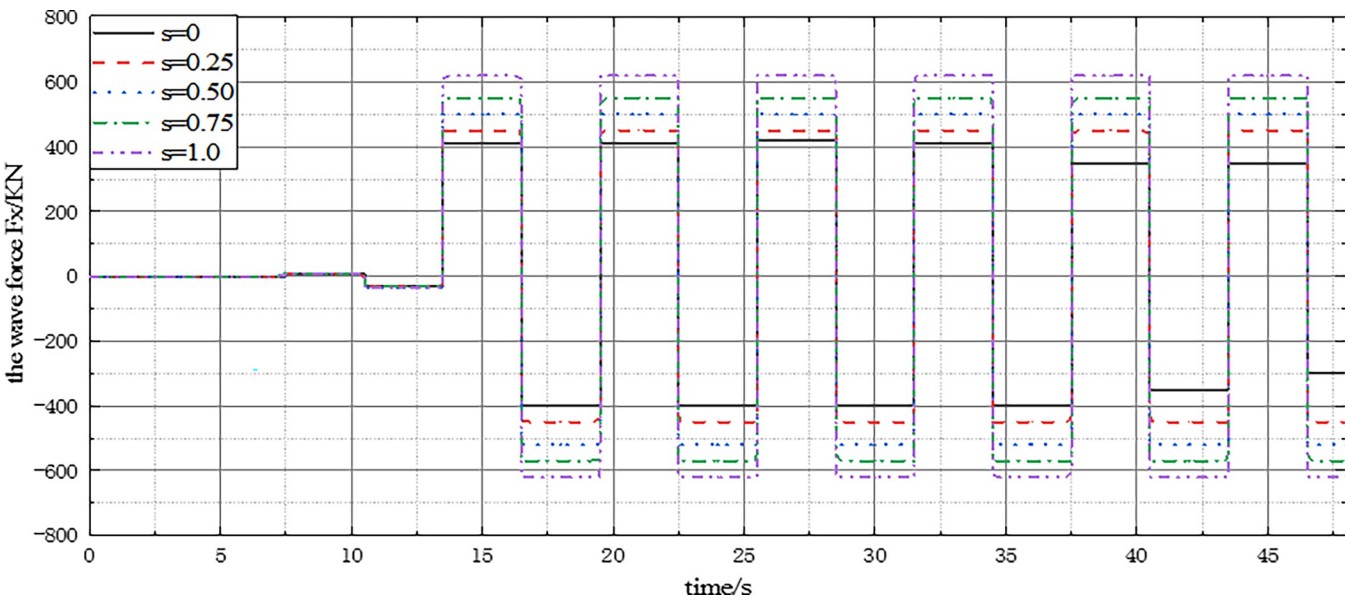

**Fig 17. $F_X$ and $F_{Xmax}$ variation curves under different buried depth s.**

foundation significantly affects its dynamic response. As the buried depth increases, the deformation of the structure gradually decreases, and the stable area at the bottom of the structure increases. The deformation at the top of the structure also decreases, and the truss between the eight-legged columns gradually shifts from torsional deformation to a normal stress state.

Moreover, the characteristics of the elastic seabed, particularly the seepage inside the seabed, significantly influence the dynamic characteristics of the structure (as shown in Fig 16). The depth of the shallow-buried structures into the seabed is relatively shallow, and hence the elastic seabed has different effects on the stress of the shallow-buried structures, depending on the buried depth of the liners.

The variation curve of FX and $FX_{max}$ under the condition of different relative burial depths presented in Fig 17 shows that as the buried depth S increases, the change period of FX remains the same, and $FX_{max}$ increases linearly with the increase of buried depth. This is because the stress calculation length of shallow buried structures increases linearly. Therefore, in the consolidation design of structures on an elastic seabed, the influence of seabed characteristics and the influence of lining buried depth on structural stress should be taken into consideration.

## 4.3 The stability of the eight-legged temporary structure under different wave parameter conditions

According to the functional design requirements of the offshore drilling platform of Long Yuan Group, specific parameters are shown in Table 1. Fig 18 is a contour plot of the deformation of the octopus supporting structure and pile foundation under different wave conditions. The octopus supporting structure deformation is analyzed separately and it can be observed that with increasing wave force, the structure undergoes significant deformation and stress concentration occurs in the rock foundation behind the main pile. The truss between the octopus' structures also experiences significant deformation, especially as shown in Fig 10A. When the wave force exceeds the horizontal bearing capacity of the pile, the deformation of the octopus-shaped retaining structure is analyzed. The front truss of the eight-legged structure along

**Table 1. Numerical analysis parameters.**

| Sea wave parameters | numerical value |
|---|---|
| Wave height $H_0$/m | 5.0 |
| Water depth $d_w$/m | 20.0 |
| Period $T$/s | 6.0 |
| Seabed parameters | numerical value |
| Seabed thickness $d_s$/m | 15.0 |
| Porosity $n$ | 0.4 |
| Roughness factor $\beta$ | 0.04 |
| Parameters of single pile | numerical value |
| Diameter of single pile $D$/m | 10 |
| Length of pile $Lm$/m | 25.0 |
| Relative buried depth $e$ | 2.0 |

the wave's advancing direction loses its stress support due to deformation. Although the truss behind the main pile tube has no deformation, its supporting force is minimal. This is because the main pile bears most of the wave force, and the rear truss can provide supporting function. Therefore, under the action of wind and waves exceeding the horizontal bearing capacity, the supporting effect of the octopus-type retaining structure is small, and the main bearing structure is the main bearing pillar.

Fig 19 depicts a visual representation of the deformation of the eight-legged supporting structure and its surrounding environment under different wave conditions. Analysis of Fig 11B and 11C) indicates that as wave forces increase, the pile foundation expands gradually from the main pile barrel to the bottom of the eight-legged support structure, creating a bowl-shaped deformation area at the bottom that gradually extends from the central pile to the eight-legged structure. It is noteworthy that Fig 19A) shows that when the wave height ratio H0/l reaches 0.16, eddy forces are generated in the seawater around the octagonal structure. Consequently, the rock base at the bottom deforms significantly, and in severe cases, the rock base may even become hollowed out.

# 5 Conclusion

This paper focuses on the stability analysis of a multi-leg support structure using the finite element simulation software MIDAS-3D. The study begins by comparing the force exerted on

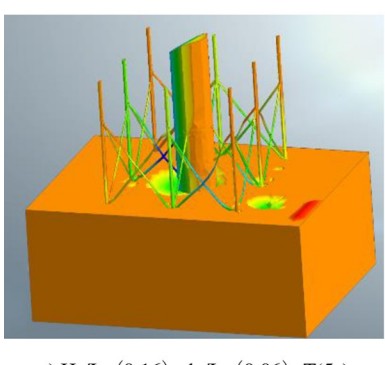 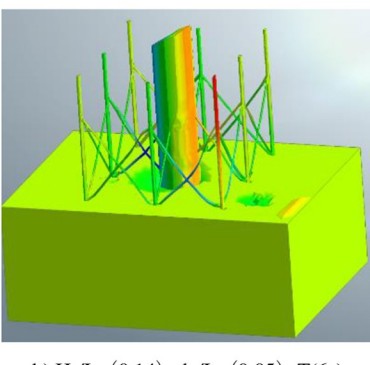 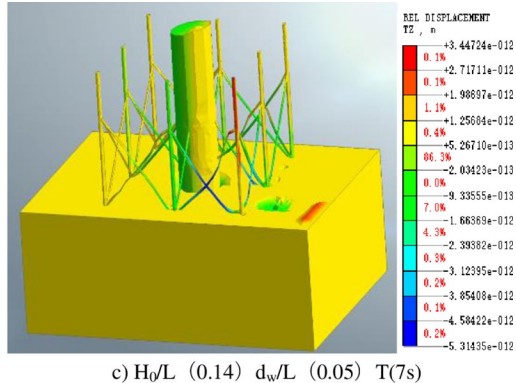

a) H$_0$/L （0.16）d$_w$/L （0.06）T(5s) b) H$_0$/L （0.14）d$_w$/L （0.05）T(6s) c) H$_0$/L （0.14）d$_w$/L （0.05）T(7s)

**Fig 18. Cloud diagram of deformation distribution of eight-leg supporting structure.**

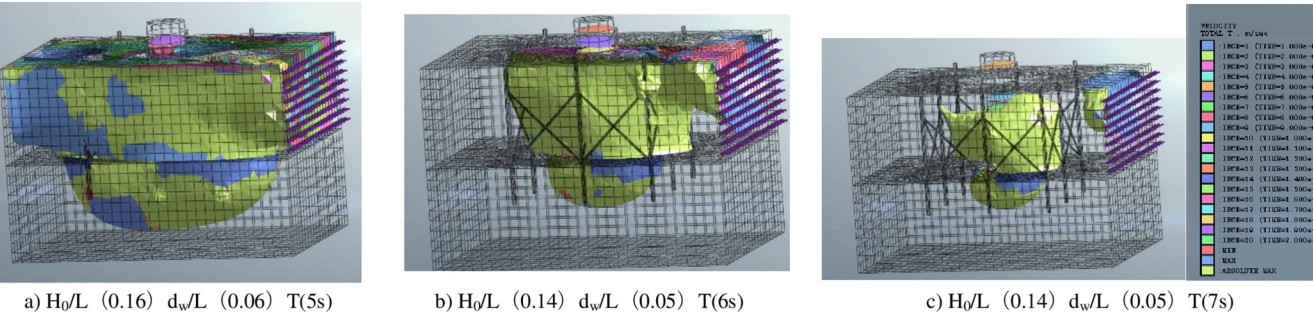

a) $H_0/L$ (0.16) $d_w/L$ (0.06) T(5s)　　b) $H_0/L$ (0.14) $d_w/L$ (0.05) T(6s)　　c) $H_0/L$ (0.14) $d_w/L$ (0.05) T(7s)

**Fig 19. Deformation rate diagram of unit under different wave parameters.**

the structure under wave load with that of the rock foundation, based on existing literature. A numerical simulation model is then employed to analyze the deformation characteristics of the eight-legged support structure under various wave and pile diameter parameters. Further-more, a detailed analysis of the deformation behavior of the octagonal support structure and rock foundation is conducted.

The findings indicate that a three-dimensional coupled numerical model considering waves in shallow water can effectively capture the wave forces on buried offshore structures. The eight-leg support structure is capable of intercepting and diverting water flow to mitigate its impact on the main pile during large wave conditions. However, an increase in the structure's diameter under the same wave conditions results in a worsened stress environment for the main pile and a decreased supporting effect of the eight-legged support structure. Numerical calculations demonstrate that the shallow pile foundation experiences significant deformation, while the eight-leg support structure continues to be supported by the dead weight of the main pile.

In addition to the stability analysis of shallow-buried offshore drilling platforms under the geological conditions of a rock seabed in the South China Sea, this paper provides recommen-dations for future research. It is suggested that future work should include stability calculations for shallow water drilling platforms under different seabed conditions. By exploring the stabil-ity behavior of drilling platforms in diverse geological contexts, a more comprehensive under-standing of their performance and design considerations can be achieved.

Overall, this study contributes valuable insights into the stability analysis of multi-leg sup-port structures under wave action, highlighting the importance of considering wave forces and their effects on the structural response. The findings and recommendations presented in this paper can serve as a basis for further research in this field.

## Acknowledgments

We thank Henan Disaster Prevention and Mitigation Engineering Center for providing equip-ment support for this experiment.

## Author Contributions

**Project administration:** Weikuan Wang.

**Software:** Yan-zhao Yuan.

**Supervision:** Weikuan Wang.

**Writing – original draft:** Yan-zhao Yuan.

**Writing – review & editing:** Yan-zhao Yuan.

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
