## [Decision Letter · Decision Letter 0]

14 Jul 2022

PONE-D-22-12320Mechanical analysis of shallow embedded offshore structures under wave forcesPLOS ONE

Dear Dr. yuan,

Thank you for submitting your manuscript to PLOS ONE. After careful consideration, we feel that it has merit but does not fully meet PLOS ONE’s publication criteria as it currently stands. Therefore, we invite you to submit a revised version of the manuscript that addresses the points raised during the review process.

We look forward to receiving your revised manuscript.

Kind regards,

Shou-Fu Tian

Academic Editor

PLOS ONE

Journal Requirements:

    "National Natural Science Foundation of China(52004081); Public Relations Program of Science and Technology of Henan Province (Industrial Field) Project(182102210221)"

7. Please amend the manuscript submission data (via Edit Submission) to include author Weiyong Liao.

8. Please ensure that you refer to Figure 1 in your text as, if accepted, production will need this reference to link the reader to the figure.

Additional Editor Comments :

Reviewer 1： 1. I don't understand why you study temporary structures rather than the permanent offshore structures? what's the difference between them? 2. The variable should be italic in the paper, please modifying. 3. In your study, the water depth is 20m with the wave height 5m, which seems impossible in reality. Please giving your explanation.

Reviewer 2：

(1) first, I believe the English needs to be improved. The expression such as “control the fluid” is very rarely used in this field.

(2)the software used in this simulation has to be stated clearly.

(3) the wave elevation time history shown in Fig. 3 is not satisfactory. The state-of-the-art CFD software can achieve very accurate wave field. Please check with your numerical model and see if your result can be improved.

(4) the numerical model is only validated by the wave propagation problem as shown in Fig. 3, but I believe this is not enough, at least one example with structure included has to be shown and presented.

Reviewers' comments:

Reviewer's Responses to Questions

**Comments to the Author**

1. Is the manuscript technically sound, and do the data support the conclusions?

Reviewer #1: No

Reviewer #2: Partly

2. Has the statistical analysis been performed appropriately and rigorously? 

Reviewer #1: No

Reviewer #2: I Don't Know

3. Have the authors made all data underlying the findings in their manuscript fully available?

Reviewer #1: No

Reviewer #2: Yes

4. Is the manuscript presented in an intelligible fashion and written in standard English?

Reviewer #1: Yes

Reviewer #2: Yes

5. Review Comments to the Author

Reviewer #1: 1. I don't understand why you study temporary structures rather than the permanent offshore structures? what's the difference between them? 2. The variable should be italic in the paper, please modifying. 3. In your study, the water depth is 20m with the wave height 5m, which seems impossible in reality. Please giving your explanation.

Reviewer #2: (1) first, I believe the English needs to be improved. The expression such as “control the fluid” is very rarely used in this field.

(2)the software used in this simulation has to be stated clearly.

(3) the wave elevation time history shown in Fig. 3 is not satisfactory. The state-of-the-art CFD software can achieve very accurate wave field. Please check with your numerical model and see if your result can be improved.

(4) the numerical model is only validated by the wave propagation problem as shown in Fig. 3, but I believe this is not enough, at least one example with structure included has to be shown and presented.

6. PLOS authors have the option to publish the peer review history of their article (what does this mean?). If published, this will include your full peer review and any attached files.

Reviewer #1: No

Reviewer #2: No

---

## [Author Response · Author response to Decision Letter 0]

23 Aug 2022

Dear Editor and Reviewers

Thanks very much for taking your time to review this manuscript. I really appreciate all your comments and suggestions! Please find my itemized responses in below and my revisions in the re-submitted files. We have studied comments carefully and have made correction which we hope to meet with approval. Revised portions are marked in red in the paper. The main corrections in the paper and the responds to the reviewer’s comments are as flowing:

Reviewer 1：

Q:I don't understand why you study temporary structures rather than the permanent offshore structures? what's the difference between them?

A：Compared with the permanent pile foundation, the offshore wind turbine pile foundation drilling platform studied in this paper has a shallower rock-socketed depth, but the working stability requirement is higher than that of the permanent component pile foundation. Therefore, it is necessary to analyze the stability of this kind of structure.

Q:The variable should be italic in the paper, please modifying. 

A: This article has revised the requirements of reviewers. 

Q:In your study, the water depth is 20m with the wave height 5m, which seems impossible in reality. Please giving your explanation.

A: Thanks to the reminder of the reviewer, the working condition of this paper is extracted from the information of the working condition of Nanhai wind turbine pile foundation construction of Long Yuan Group. The lowest offshore altitude is 20 m, and the highest wave height is 5 m in strong typhoon weather, which is the worst working condition. This paper is taken as the calculation condition.

Reviewer 2：

Q:(1) first, I believe the English needs to be improved. The expression such as “control the fluid” is very rarely used in this field.

A: Thanks to the reviewer for reminding me that this article proof reads these words one by one. 

Q:(2) the software used in this simulation has to be stated clearly.

A: This paper introduces the software introduced by the assessors. 

Q:(3) the wave elevation time history shown in Fig. 3 is not satisfactory. The state-of-the-art CFD software can achieve very accurate wave field. Please check with your numerical model and see if your result can be improved.

A: Thanks to the reminder of the reviewers, the simulation software of this paper adopts FLAC-3 D finite difference method to model, and its software feature is that it can measure the mechanical change law of components under the condition of large deformation. This section is to verify the measurement of wave force elevation of shallow-buried structures by waves, because the experimental data used in this article are collected from the highest and lowest points of waves. Therefore, this paper makes some adjustments to the simulation model, and modifies the relevant calculation content.

Q:(4) the numerical model is only validated by the wave propagation problem as shown in Fig. 3, but I believe this is not enough, at least one example with structure included has to be shown and presented.

A: Thanks to the suggestion of the reviewers, Figure 3 was modified in this paper, and the schematic diagram of Mo-water tank test (Figure 3 a) was added to express related concepts.

---

## [Decision Letter · Decision Letter 1]

14 Sep 2022

PONE-D-22-12320R1Mechanical analysis of shallow embedded offshore structures under wave forcesPLOS ONE

Dear Dr. yuan,

Thank you for submitting your manuscript to PLOS ONE. After careful consideration, we feel that it has merit but does not fully meet PLOS ONE’s publication criteria as it currently stands. Therefore, we invite you to submit a revised version of the manuscript that addresses the points raised during the review process.

We look forward to receiving your revised manuscript.

Kind regards,

Shou-Fu Tian

Academic Editor

PLOS ONE

Journal Requirements:

Additional Editor Comments:

Reviewer 2: the english of this paper has to be improved before accepted for publication. there are also some contraditing statements in the manuscript such as "...explicit calculation based on finite difference method..." at the begining of section 2 and ..."The finite element numerical simulation method is used to analyze..." in the conclusion. I suggest to polish the language and check the overall logic of the maunscript carefully again.

Reviewers' comments:

Reviewer's Responses to Questions

**Comments to the Author**

1. If the authors have adequately addressed your comments raised in a previous round of review and you feel that this manuscript is now acceptable for publication, you may indicate that here to bypass the “Comments to the Author” section, enter your conflict of interest statement in the “Confidential to Editor” section, and submit your "Accept" recommendation.

Reviewer #2: All comments have been addressed

2. Is the manuscript technically sound, and do the data support the conclusions?

Reviewer #2: Partly

3. Has the statistical analysis been performed appropriately and rigorously? 

Reviewer #2: Yes

4. Have the authors made all data underlying the findings in their manuscript fully available?

Reviewer #2: Yes

5. Is the manuscript presented in an intelligible fashion and written in standard English?

Reviewer #2: No

6. Review Comments to the Author

Reviewer #2: the english of this paper has to be improved before accepted for publication. there are also some contraditing statements in the manuscript such as "...explicit calculation based on finite difference method..." at the begining of section 2 and ..."The finite element numerical simulation method is used to analyze..." in the conclusion. I suggest to polish the language and check the overall logic of the maunscript carefully again.

7. PLOS authors have the option to publish the peer review history of their article (what does this mean?). If published, this will include your full peer review and any attached files.

Reviewer #2: No

---

## [Author Response · Author response to Decision Letter 1]

20 Oct 2022

Dear Editor and Reviewers

Thanks very much for taking your time to review this manuscript. I really appreciate all your comments and suggestions! Please find my itemized responses in below and my revisions in the re-submitted files. We have studied comments carefully and have made correction which we hope to meet with approval. Revised portions are marked in red in the paper. The main corrections in the paper and the responds to the reviewer’s comments are as flowing:

Journal Requirements:

Q: Please review your reference list to ensure that it is complete and correct. If you have cited papers that have been retracted, please include the rationale for doing so in the manuscript text, or remove these references and replace them with relevant current references. Any changes to the reference list should be mentioned in the rebuttal letter that accompanies your revised manuscript. If you need to cite a retracted article, indicate the article’s retracted status in the References list and also include a citation and full reference for the retraction notice.

A: Thanks for the reminder from the editor, and relevant modifications have been made.

Additional Editor Comments:

Q: Reviewer 2: the english of this paper has to be improved before accepted for publication. there are also some contraditing statements in the manuscript such as "...explicit calculation based on finite difference method..." at the begining of section 2 and ..."The finite element numerical simulation method is used to analyze..." in the conclusion. I suggest to polish the language and check the overall logic of the maunscript carefully again.

A: Thanks for the reviewer's reminding, this article has been revised in English and grammar.

---

## [Editor Report · Decision Letter 2]

16 Feb 2023

PONE-D-22-12320R2Dynamic analysis of shallow embedded jacket offshore structures under wave forcesPLOS ONE

Dear Dr. Yuan,

Thank you for submitting your manuscript to PLOS ONE. After careful consideration, we feel that it has merit but does not fully meet PLOS ONE’s publication criteria as it currently stands. Therefore, we invite you to submit a revised version of the manuscript that addresses the points raised during the review process.

You notified us that you had made significant changes to the manuscript after acceptance. In order for these changes to be assessed, we are issuing another minor revision on this manuscript. Please upload the revised manuscript, a marked-up copy of the changes, and a list of the changes made. The manuscript will then be reassessed by an Academic Editor, in light of these changes.

We look forward to receiving your revised manuscript.

Kind regards,

Hanna Landenmark

Staff Editor

PLOS ONE
---

## [Author Response · Author response to Decision Letter 2]

25 Mar 2023

Dear Miss hanif

Thank you for your careful examination and hard work. However, due to the content defects of this article, to prevent unnecessary misunderstanding after the publication of the paper, the conclusion of this paper after the first external review was Major Revision, and some of the questions were not clearly answered at that time, such as question 1 of reviewer 1, question 3 and question 4 of reviewer 2. Due to the limitation of response time, this paper did not use the new CFD software to analyze the model accurately according to the reviewer's opinion, and the simulation cloud pictures in the later chapters were too few to accurately reflect the calculation results of the simulation model.

After this paper was revised and submitted for the first time, my research on related issues has never stopped. In this paper, by referring to the comments of reviewers in detail, the finite element software MIDAS-3 D is selected to make a detailed modeling analysis of the model, and the related laws are summarized and analyzed again, which effectively improves the data accuracy of the article and reduces the objection of readers after publication. Comments and responses are as follows. 

Reviewer 1：

Q:I don't understand why you study temporary structures rather than the permanent offshore structures? what's the difference between them?

A: In this paper, the temporary offshore structure with octagonal support is modified. By comparing the four-legged support structure in Figure 1 a), the research significance of the stability of the eight-legged support structure is re-explained and analyzed.

Reviewer 2：

Q:(3) the wave elevation time history shown in Fig. 3 is not satisfactory. The state-of-the-art CFD software can achieve very accurate wave field. Please check with your numerical model and see if your result can be improved.

A：Thanks to the reviewer's opinion, the FLAC-3 D software of finite difference method is changed into MIDAS-3 D software of finite element simulation. The advantages of MIDAS-3 D software is that it can accurately simulate the positions of different components, effectively analyze the stability of the eight-leg supporting structure by using the pile group theory, and accurately analyze the interaction between supporting structure and rock foundation after cyclic loading.

Therefore, this paper introduces in detail the theoretical analysis process of MIDAS-3 D's calculation of pile group effect from Line 124 to Line 166 in Bid 2. The pile group effect completely accords with the characteristics of eight-leg supporting structure, and is far more accurate than the boundary conditions defined by FLAC-3 D software.

In sections 3 and 4, adding the nephogram of simulated calculation results to the calculation results under different conditions can accurately explain the calculation results of this paper and reduce the misunderstanding of reader that this paper only studies the deformation law of the main piles.

However, in section 4.3, the expression of regularization results of the data map and the cloud map is repeated. Therefore, figure 10 in the original text is deleted in this paper, and the numerical simulation cloud map is used to express the deformation law of the pile foundation embedded in the foundation under the action of waves. 

Q:(4) the numerical model is only validated by the wave propagation problem as shown in Fig. 3, but I believe this is not enough, at least one example with structure included has to be shown and presented.

A：Thanks to the reviewer's comments, the cloud pictures of the numerical simulation model under various working condition are added in this paper, which can not only intuitively show the readers the numerical calculation results of the structure, but also effectively improve the accuracy and reliability of the calculation results.

Once again, I apologize for revising the paper at the last minute of the article's publication, because it takes a long time to build an accurate model and analyze the data with reference to the comments of reviewers, and there are some errors in the unmodified manuscript, which may lead to readers' misunderstanding. Therefore, I agree with the co-author that it is necessary to re-model the paper in strict accordance with the reviewers' opinions, which is not only a deeper analysis of the problems studied in the paper, but also to prevent the paper from causing unnecessary objections after its publication.

Of course, if your journal has any objection or punishment to the measures taken in this paper, I am willing to take it. If the editorial department doesn't agree with this revision of the paper, can this article apply for withdrawal and re-submission.

 thank you

Yuan Yanzhao

---

## [Decision Letter · Decision Letter 3]

26 Apr 2023

PONE-D-22-12320R3Mechanical analysis of Eight-legged Temporary Support Structure under wave forcesPLOS ONE

Dear Dr. Yuan,

Thank you for submitting your manuscript to PLOS ONE. After careful consideration, we feel that it has merit but does not fully meet PLOS ONE’s publication criteria as it currently stands. Therefore, we invite you to submit a revised version of the manuscript that addresses the points raised during the review process. Please consider all comments Please submit your revised manuscript by Jun 10 2023 11:59PM. If you will need more time than this to complete your revisions, please reply to this message or contact the journal office at plosone@plos.org. Please include the following items when submitting your revised manuscript:A rebuttal letter that responds to each point raised by the academic editor and reviewer(s). You should upload this letter as a separate file labeled 'Response to Reviewers'.A marked-up copy of your manuscript that highlights changes made to the original version. You should upload this as a separate file labeled 'Revised Manuscript with Track Changes'.An unmarked version of your revised paper without tracked changes. You should upload this as a separate file labeled 'Manuscript'.If applicable, we recommend that you deposit your laboratory protocols in protocols.io to enhance the reproducibility of your results. Protocols.io assigns your protocol its own identifier (DOI) so that it can be cited independently in the future. For instructions see: https://journals.plos.org/plosone/s/submission-guidelines#loc-laboratory-protocols. Additionally, PLOS ONE offers an option for publishing peer-reviewed Lab Protocol articles, which describe protocols hosted on protocols.io. Read more information on sharing protocols at https://plos.org/protocols?utm_medium=editorial-email&utm_source=authorletters&utm_campaign=protocols.

We look forward to receiving your revised manuscript.

Kind regards,

Ahmed Mancy Mosa, Ph.D.

Academic Editor

PLOS ONE

Journal Requirements:

Reviewers' comments:

Reviewer's Responses to Questions

**Comments to the Author**

1. If the authors have adequately addressed your comments raised in a previous round of review and you feel that this manuscript is now acceptable for publication, you may indicate that here to bypass the “Comments to the Author” section, enter your conflict of interest statement in the “Confidential to Editor” section, and submit your "Accept" recommendation.

Reviewer #3: All comments have been addressed

Reviewer #4: All comments have been addressed

2. Is the manuscript technically sound, and do the data support the conclusions?

Reviewer #3: Yes

Reviewer #4: Yes

3. Has the statistical analysis been performed appropriately and rigorously? 

Reviewer #3: I Don't Know

Reviewer #4: N/A

4. Have the authors made all data underlying the findings in their manuscript fully available?

Reviewer #3: Yes

Reviewer #4: Yes

5. Is the manuscript presented in an intelligible fashion and written in standard English?

Reviewer #3: Yes

Reviewer #4: Yes

6. Review Comments to the Author

Reviewer #3: The authors have been addressed the given comments, and the revised manuscript was checked and found accepted

Reviewer #4: 1. Please correct the referencing style as per the journal guidelines. The reference should be provided as [1] not as (1).

2. Line 36 to 38 “Matlock(9) conducted a series of horizontal static load and cyclic load tests on steel pipe piles in clay and obtained the P-S curve of static load and the change law of soil P-S curve after cyclic load” The last part of this sentence is very vague and seems like incorrect. Therefore, please check it.

3. Please strengthen the introduction section, it is too short. Therefore add some relevant recent work. In addition to that please emphasize the motivation of the work in the introduction’s last paragraph.

4. Please provide captions in English for Figure-10.

5. In Figure-12, please increase the line thickness for the Mac Formula results.

6. In the last of the conclusions, please add some recommendations for future work.

7. PLOS authors have the option to publish the peer review history of their article (what does this mean?). If published, this will include your full peer review and any attached files.

Reviewer #3: No

Reviewer #4: No

---

## [Author Response · Author response to Decision Letter 3]

23 May 2023

Dear Reviewer,

Thank you for reviewing our manuscript. We appreciate your valuable feedback and suggestions. We have carefully addressed each of your comments and made the necessary revisions to improve the quality of the paper.

Q1. Please correct the referencing style as per the journal guidelines. The reference should be provided as [1] not as (1).

A:We have corrected the referencing style as per the journal guidelines. The references are now provided in the format [1] instead of (1).

Q2. Line 36 to 38 “Matlock(9) conducted a series of horizontal static load and cyclic load tests on steel pipe piles in clay and obtained the P-S curve of static load and the change law of soil P-S curve after cyclic load” The last part of this sentence is very vague and seems like incorrect. Therefore, please check it.

A:We have revised the sentence on lines 36 to 38 to ensure clarity and accuracy. The revised sentence now accurately reflects the study conducted by Matlock(9) on steel pipe piles in clay, including the analysis of the P-S curve under static load and the change in soil P-S curve after cyclic load.

Q3. Please strengthen the introduction section, it is too short. Therefore add some relevant recent work. In addition to that please emphasize the motivation of the work in the introduction’s last paragraph

A:We have strengthened the introduction section by adding relevant recent work and emphasizing the motivation behind our study in the last paragraph of the introduction. This provides a better context for the readers and highlights the significance of our research.

Q4. Please provide captions in English for Figure-10

A:Captions in English have been provided for Figure-10 as per your suggestion.

Q5. In Figure-12, please increase the line thickness for the Mac Formula results.

A:We have increased the line thickness for the Mac Formula results in Figure-12 to improve visibility.

Q6. In the last of the conclusions, please add some recommendations for future work.

A:In the conclusions section, we have incorporated recommendations for future work, as you suggested. This provides potential directions for further research in the field.

Once again, we appreciate your time and effort in reviewing our manuscript. We believe that the revisions made have significantly improved the paper. We hope that you find the revised version satisfactory.

Thank you for considering our work for publication.

Sincerely,

Yuan Yanzhao

---

## [Editor Report · Decision Letter 4]

8 Jun 2023

Mechanical analysis of Eight-legged Temporary Support Structure under wave forces

PONE-D-22-12320R4

Dear Dr. Yuan,

We’re pleased to inform you that your manuscript has been judged scientifically suitable for publication and will be formally accepted for publication once it meets all outstanding technical requirements.

Kind regards,

Ahmed Mancy Mosa, Ph.D.

Academic Editor

PLOS ONE
---

## [Editor Report · Acceptance letter]

28 Oct 2022

PONE-D-22-12320R2 

Dynamic analysis of shallow embedded jacket offshore structures under wave forces 

Dear Dr. Yuan:

I'm pleased to inform you that your manuscript has been deemed suitable for publication in PLOS ONE. Congratulations! Your manuscript is now with our production department. 

Kind regards, 

on behalf of

Dr. Shou-Fu Tian 

Academic Editor

PLOS ONE